# Wellbeing Literacy: Conceptualization, Measurement, and Preliminary Empirical Findings from Students, Parents and School Staff

**DOI:** 10.3390/ijerph18041485

**Published:** 2021-02-04

**Authors:** Hanchao Hou, Tan-Chyuan Chin, Gavin R. Slemp, Lindsay G. Oades

**Affiliations:** Centre for Positive Psychology, Melbourne Graduate School of Education, University of Melbourne, Parkville, VIC 3010, Australia; tanchyuan.chin@unimelb.edu.au (T.-C.C.); gavin.slemp@unimelb.edu.au (G.R.S.); lindsay.oades@unimelb.edu.au (L.G.O.)

**Keywords:** wellbeing literacy, wellbeing, public health, educational context, measurement

## Abstract

Aims: Wellbeing literacy is the intentional use of wellbeing relevant vocabulary, knowledge and language skills to maintain or improve the wellbeing of oneself, others and the world. In this study, we operationalize the human aspects of the concept of wellbeing literacy and empirically test its relationship with wellbeing and illbeing. We also assess its incremental variance in wellbeing and illbeing, after controlling for existing and well-established predictors of these constructs within education settings. Methods: We developed and empirically tested the Wellbeing literacy 6-item (Well-Lit 6) scale to assess the concept of wellbeing literacy in the education context. The scale was developed based on a working definition of wellbeing literacy, in combination with the Australian Curriculum, Assessment and Reporting Authority (ACARA)’s definition of literacy. The Well-Lit 6 was administered via a cross-sectional survey to three Australian samples that comprise different elements of Australian education systems: students (*N* = 1392), parents (*N* = 584) and school staff (*N* = 317). Results: Exploratory factor analysis (EFA) suggested the six items of the Well-Lit 6 form an independent construct, empirically distinguishable from other wellbeing-related constructs (e.g., general wellbeing, resilience, and emotion regulation). Convergent analyses showed wellbeing literacy was positively related to wellbeing and negatively related to illbeing. Incremental validity analyses showed wellbeing literacy predicted variance in wellbeing and illbeing after controlling for participant demographics, resilience, and emotion regulation, showing initial evidence of incremental validity. Conclusions: Our results provide preliminary evidence that wellbeing literacy is a distinct construct from wellbeing and illbeing, and it also demonstrates significant unique variance in these constructs over and above resilience and emotion regulation. The Well-Lit 6 is a useful provisional measure of wellbeing literacy, although we suggest a fruitful avenue for future research is to develop a more comprehensive scale of wellbeing literacy that denotes specific facets of communication, allowing a fuller exploration wellbeing literacy, its components, and their antecedents and consequences. We offer further recommendations for future research and discuss limitations with our approach.

## 1. Introduction

Wellbeing is an important concept in public health and the social sciences. The World Health Organization (WHO) defines health as “a state of complete physical, mental and social wellbeing and not merely the absence of disease or infirmity” [1]. This definition of health highlights two points: (1) health includes multiple factors, such as physical, mental and social aspects, and (2) it is more than simply the absence of negative situations in life. While there continues to be a debate about the precise definition of wellbeing, scholars generally agree about these two dimensions of public health [2,3,4]. More holistic conceptions are similarly making their way into psychology and other social sciences to conceptualize the construct of wellbeing [5,6]. In the present study, we adopt this view of wellbeing, suggesting it is not just the absence of illness, but also encompasses the presence of positive qualities. In doing so, we propose what we consider to be an important capability-based antecedent of wellbeing, which we term wellbeing literacy.

### 1.1. Health Literacies, Education, and Public Health

Education is a systematic way to improve public health. It allows for nourishing psychosocial environments that support human development (e.g., sense of control and social support), work (e.g., working conditions and income), and helps to foster health knowledge and behaviours [7]. There are currently two main initiatives that address the interconnection between health and education: health literacy and mental health literacy. 

Health literacy is defined as “the cognitive and social skills which determine the motivation and ability of individuals to gain access to, understand and use information in ways which promote and maintain good health” [8] (p. 357). Focusing primarily on physical health, Nutbeam proposed health literacy as the achievement of sufficient health knowledge, as well as skills and confidence to improve personal and community health by changing personal living conditions and lifestyles [8]. Examples of health literacy include knowledge and maintenance of a healthy diet, understanding how to prevent skin cancer, first-aid skills, and understanding how to access further health information. Thus, the construct itself is dependent on adequate levels of literacy, with poor literacy limiting one’s personal, social, and cultural development.

Definitions and measurement tools of health literacy [9,10] continue to evolve from a narrow focus on risk factors in clinical settings to more complex and broader conceptions involving multiple factors and covering the whole community [10,11,12]. However, they generally share in common a focus on physical health and evidence also suggests that health literacy is associated with health outcomes. A comprehensive systematic review, for example, concluded that low health literacy is associated with lower health outcomes, including poorer overall health status, poorer vaccination rates, and higher mortality [13]. Evidence such as this supports the effort to incorporate health literacy into education.

Mental health literacy was proposed as a related yet complementary construct to health literacy, differing insofar as it calls people’s attention to mental, rather than physical, health [14,15]. The definition of mental health literacy is “knowledge and beliefs about mental disorders, which aid their recognition, management, or prevention” [15] (p. 396). Several dimensions have been proposed, including the ability to recognize disorders or different types of illbeing, knowledge and beliefs about mental illness risk factors and causes, knowledge about self-help interventions, professional help, or how to seek help, and attitudes that facilitate recognition and appropriate help-seeking behavior [14]. Hence, like health literacy, mental health literacy requires adequate levels of literacy, for successful navigation, comprehension, and effective use of relevant mental health disorder information.

Several kinds of mental health literacy interventions have been implemented in schools and communities to decrease the prevalence of mental illness [16], with some evidence suggesting mental health literacy helps improve mental health, or mental illness prevention. For instance, evidence from a systematic review of online mental health literacy interventions [17] shows its efficacy in improving help-seeking behaviour and uptake of mental health services, and in reducing depression. While promising, a limitation of mental health literacy is an almost exclusive focus on mental illness, with theorists and researchers remaining silent about the positive aspects of mental health. The broader role of how language and its use impacts on the way health and wellbeing is experienced is also overlooked. We assert that mental health literacy and health literacy may actually be knowledge, and not literacy at all. Thus, we introduce the concept of wellbeing literacy to address this gap.

### 1.2. Wellbeing Literacy

Wellbeing literacy is defined as vocabulary, knowledge and language skills that may be intentionally used to maintain or improve the wellbeing of oneself or others [18,19]. Wellbeing literacy is concerned with how and why people use language in their everyday lives, and how the use of language can be leveraged to produce wellbeing for the self or others. With a greater capacity for language-use choices that can benefit their own wellbeing and that of others, people who have wellbeing literacy thus possess a capability for wellbeing [20,21]. Oades et al. explicate five dimensions of wellbeing literacy that reflect its necessary conditions and that are used in its operationalization [19]. First, wellbeing literacy requires vocabulary and knowledge. Proficiency in vocabulary ensures people can articulate ways wellbeing can be enhanced for the self or others (e.g., ability to clearly express things that matter to oneself). Knowledge ensures that language is “about something”, relevant to the person. Second, wellbeing literacy requires a capacity to comprehend wellbeing related communication via literacy-based tools such as reading, listening, or viewing [22]. This might involve, for example, reading about different emotions so that one can more accurately and reliability discern different types of emotional experiences. Third, wellbeing literacy requires the capacity to compose wellbeing relevant “texts” across multiple modalities: writing, speaking, or creating [22]. For example, just as a proficient science communicator adopts audience-appropriate language to meaningfully convey technical-scientific research, effective wellbeing-related information requires composition that is appropriate for specific value-systems and social contexts. Fourth, because wellbeing is conceptually bound [23], wellbeing literacy requires careful attention paid to the context, so that adaptations of wellbeing communications can be made. The meaning of language changes with context, to which a wellbeing literate person will adapt. For example, the meaning of “wellbeing” itself is culturally variant and will thus involve different interpretations across settings with vastly different cultural roots and language systems [24,25]. Finally, wellbeing literacy requires wellbeing intentionality [26], involving a desire to deliberately use language to improve the wellbeing of oneself, others or the world. Intentionality here also refers to the idea of a “habit of intention”. That is, the person has the choice to use the language skills and ability on multiple occasions, not a single instance of one intention. The Buddhist notion of “right speech” as part of the eightfold path is a similar concept [27]. That is, using the speech is seen as important morally due to its impact on the mind and others. Likewise, wellbeing literacy includes the right moral intentions as part of its conceptual definition, recognising that language does not use itself, rather it has a user with intentions.

Wellbeing literacy holds promise as a means to enhance wellbeing because it is not an intervention per se, but rather is a language-based capability; that is, a repertoire of skills and options one has available to influence their wellbeing. The notion of building capabilities is conducive to the growth-based emphasis of holistic conceptions of wellbeing, such as that proposed by WHO, that comprise both the absence of illness as well as the presence of positive mental health. Moreover, unlike mental health literacy, wellbeing literacy shifts the focus to what is possible, rather than what is broken. Despite growing conceptual arguments for wellbeing literacy [18,28,29,30,31], few published research studies have directly examined this new construct, likely due to the lack of an available scale to reliably and validly measure it. Thus, little is known about its antecedents, consequences, or its relationship with wellbeing over and above existing constructs.

### 1.3. A Short Scale of Wellbeing Literacy 

To address these issues, the present study seeks to operationalize and validate a short, empirical measure of the construct of wellbeing literacy, which we have titled the Wellbeing Literacy 6 item (Well-Lit 6) scale. The scale was developed from an initial item pool to be included in broader and established batteries of psychological questionnaires that are regularly offered throughout Australia to assess wellbeing in education systems. Specifically, Well-Lit 6 was included in three batteries that are designed to uniquely examine the wellbeing of students, their parents, and school staff across Australia. Because Well-Lit 6 was included in these questionnaires, a requirement was that it was brief in order to capture the breadth of the population. Hence, the scale was developed based on the working definition of wellbeing literacy, as well as its five necessary conditions, with consideration also given to existing health literacies (health literacy and mental health literacy). This produced an item-pool of 6-items, displayed in Table 1.

As shown in the Table 1, each item of the Well-Lit 6 maps onto the necessary conditions of wellbeing literacy. The items cover wellbeing knowledge, vocabulary, skills to comprehend and compose language and to adjust language depending on context (audience). Whilst intentionality is not explicitly mentioned in the items, it remains a conceptual assumption underpinning the scale.

We assert that wellbeing literacy is a capability, and hence different from wellbeing per se, which we view as an experienced state, and subjective in nature. Wellbeing literacy as a capability involves the potential to use language in the service of wellbeing. This is an important distinction directly relevant to this study. Stated simply, some educators may assert they are “teaching wellbeing”. Rather, we claim that they are using language to enable students to learn about how to improve their wellbeing or the wellbeing of others. The potential and re-usable language used by the teacher, and hopefully the learner, constitutes wellbeing literacy, not wellbeing itself. The experience of wellbeing is exactly that—an experience—therefore the language used to teach about and for wellbeing is literacy. A cognitive evaluation of an experience over time, such as life satisfaction, is also different from wellbeing literacy, as it represents a subjective evaluation of a state, rather than the building of a capability. A wellbeing literate individual may, however, find it easier to evaluate their experiences of wellbeing. 

Because construct proliferation and construct redundancy are major problems in the social sciences today [32], an important task when evaluating measures of new constructs is to establish a nomological network by assessing convergent and discriminant relations with conceptually related constructs. It is also important to examine a new construct for incremental predictive validity. These steps, together, help researchers to avoid the “old wine in new wine-skins” phenomenon. Thus, we examine the extent to which wellbeing literacy is empirically distinguishable from indicators of wellbeing, as well as other established predictors of wellbeing. 

To do this we use three approaches. First, to establish convergent validity, we evaluate whether wellbeing literacy is related in expected directions with constructs covering related conceptual terrain. Relations should be substantive although not strong enough to suggest duplication [33]. Positive relationships with other positively oriented constructs (e.g., wellbeing, life satisfaction), and negative relationships with opposing constructs (e.g., illbeing, loneliness) shows evidence of convergent validity. Second, to test discriminant validity, we examine whether wellbeing literacy loads on its own factor when placed into a broader exploratory factor analysis model inclusive of a range of other wellbeing indicators. Finally, to examine incremental validity, we explore whether wellbeing literacy adds additive variance in wellbeing over and above established wellbeing predictors. For the present study, we focus on resilience and emotion regulation, which are commonly proposed as important antecedents of wellbeing [34,35,36,37,38]. For example, resilience has been shown to be a reliable predictor of subjective wellbeing [39] as well as illbeing [40]. Meta-analytic evidence also suggests relatively robust relationships between resilience and wellbeing [41,42] and other research suggests self-reported resilience is not just a redundant measure of wellbeing [43,44]. Similarly, emotion regulation is an established predictor of wellbeing [45,46], and also shows evidence of discriminant validity over other coping mechanisms [45]. Therefore, we examine whether wellbeing literacy explains incremental variance over and above these two constructs. As wellbeing literacy includes being sensitive to one’s context, and intentionality, it is likely that both involve an aspect of self-regulation and/or metacognition. Whilst emotional regulation is likely to share these metacognitive components with wellbeing literacy, we assert that they are different constructs.

### 1.4. Aims and Hypotheses

Although wellbeing literacy is a novel and important construct that could inform psychological and educational research, there has yet been limited endeavour to develop a quantitative measure of the construct that could be used in research. It is therefore currently unknown whether wellbeing literacy is empirically distinguishable from related constructs that are thought to represent its consequences, such as wellbeing or illbeing. Moreover, there is no existing evidence about whether wellbeing literacy will serve as an important antecedent to these outcomes. The aim of this research is to investigate the construct of wellbeing literacy in educational settings, examining the first available measure of wellbeing literacy (Well-Lit 6) and its position in the nomological network. In conducting our research, we hypothesize that:

**Hypothesis 1** **(H1):***There will be a significant positive relationship between wellbeing literacy and wellbeing, including life satisfaction and positive mental health (demonstrating convergent validity)* .

**Hypothesis 2** **(H2):***There will be a significant negative relationship between wellbeing literacy and illbeing, including depression, anxiety, stress (demonstrating convergent validity)* .

**Hypothesis 3** **(H3):***Wellbeing literacy will load on its own factor when placed into a broader exploratory factor analysis (EFA) model containing related wellbeing and illbeing constructs (demonstrating discriminant validity)* .

**Hypothesis 4** **(H4):***Wellbeing literacy will predict incremental variance in wellbeing after controlling for resilience and emotion regulation (demonstrating incremental validity)* .

## 2. Method

### 2.1. Participants and Procedure

The present study was completed across three samples of participants, including students, parents, and school staff. The student sample consisted of 1392 students, comprising 722 girls and 603 boys (67 other). Their mean age was 14.47 years (SD = 1.77). The parent sample consisted of 584 parents (505 mothers, 73 fathers, 6 other). Their mean age was 42.69 years (SD = 6.32). The sample of school staff consisted of 317 employee respondents, including 251 females and 61 males (5 other). Their mean age was 42.45 years (SD = 11.95). All participants were recruited from Victoria in Australia, and informed consent was obtained prior to participation in the study. The study was approved by the Human Research Ethics Committee at the University of Melbourne (#1648526, 1853389.1, and 1851964.1 for students, parents, and staff, respectively) and the Victorian Department of Education and Training (#2019_004002). 

### 2.2. Measures

All three samples of participants were recruited as part of a larger community-wide wellbeing project. All samples completed a survey battery comprised of wellbeing literacy, as well as different aspects of wellbeing and illbeing. The measures contained within each battery are listed in Table 2, as well as which sample completed each measure. 

As shown in Table 2, the student sample completed all wellbeing predictors, in addition to all illbeing measures and life satisfaction. The parent and staff samples completed all wellbeing measures except life satisfaction, as well as the emotion regulation and resilience wellbeing predictors. The parents and school staff took fewer measures than students because (1) some items in the survey battery are designed for students, and (2) there was less available time for parents and staff to do the measurements. The Well-Lit 6 was measured in all samples. A more detailed description of the scales is provided below.

#### 2.2.1. Wellbeing Literacy

The wellbeing literacy 6-item (Well-Lit 6) scale was developed from the working definition that wellbeing literacy is vocabulary, knowledge and language skills that may be intentionally used to maintain or improve the wellbeing of oneself or others [18], and was adopted from the Australian Curriculum, Assessment and Reporting Authority (ACARA)’s definition of literacy [22]. The items and their interpretation are shown in Table 1, which is also mapped onto the necessary conditions of wellbeing literacy. It is rated on a 7-point Likert scale, from 1 “Strongly disagree” to 7 “Strongly agree”. The items showed high internal consistent reliability in all samples (students: α = 0.84, staff: α = 0.91, parents: α = 0.91).

#### 2.2.2. Wellbeing 

We measured wellbeing with the Mental Health Continuum-Short Form (MHC-SF) [47]. The MHC-SF is a widely used 14-item, self-report questionnaire assessing emotional wellbeing (3 items; sample: “In the past month, how often did you feel happy?”), psychological wellbeing (6 items; sample: “In the past month, how often did you feel good at managing the responsibilities of your daily life?”), and social wellbeing (5 items; sample: “In the past month, how often did you feel that people are basically good?”). Respondents are asked to describe the frequency with which they experience each feeling in the past month on a 6-points Likert scale from “never” to “every day”. The MHC-SF had shown good psychometric properties in previous studies [47,48,49]. In the present study, the items showed high internal consistent reliability for emotional wellbeing (staff: α = 0.89, parents: α = 0.90), psychological wellbeing (staff: α = 0.87, parents: α = 0.87), and social wellbeing (staff: α = 0.86, parents: α = 0.86).

We also assessed life satisfaction with a 2-items adapted from the Student’s Life Satisfaction Scale (SLSS) [50] in the student sample. A sample item is “My life is going well”. The items showed high internal consistent reliability (α = 0.90).

#### 2.2.3. Illbeing

We broadly measured illbeing using more specific items to evaluate Depression, Anxiety, and Stress. Depression and anxiety were both assessed with 2-items from the Patient-Health Questionnaire-4 (PHQ-4) [51]. Stress was assessed with 2-items that were adapted from the Perceived Stress Scale [52]. For all scales, participants responded using a 7-point Likert scale, from 1 “Strongly disagree” to 7 “Strongly agree”. To evaluate the social component of illbeing, we also assessed loneliness with 2 items that were contained within the survey battery for all samples. A sample item is “I often feel misunderstood and alone”. For the student sample, one more item “I don’t fit in with my peers” was added, considering the importance of peer relationships for students at school [53,54]. The scales showed tolerable to high internal consistency (α = 0.52 to 0.93).

#### 2.2.4. Wellbeing and Illbeing Predictors 

To assess the incremental validity of the Well-Lit 6 scale, we evaluated whether it showed additive variance in the wellbeing or illbeing measures over and above established predictors, including emotion regulation and resilience. To measure emotion regulation, we used a collection of items across encompassing emotion and cognitive emotion-regulation subdomains. The emotional components included reappraisal (4-items; sample: “When I want to feel less negative emotions such as sadness, I change what I’m thinking about”) and suppression (2-items; sample: “I control my emotions by not showing them to others”) and were adapted from the emotion regulation questionnaire (ERQ) [45]. We measured the cognitive component of emotion regulation using 2-items adapted from the rumination subscale of the cognitive emotion regulation questionnaire (CERQ) [55] (sample: “I often spend time re-living a past experience in my head”). The Cronbach’s alpha values for the reappraisal, rumination and suppression scales were 0.81, 0.70, and 0.69, respectively.

To measure resilience, we used 5-items adapted from the brief resilience scale (BRS) [56]. A sample item is: “I quickly get over and recover from setbacks”. The Cronbach’s alpha for the scale was 0.71.

### 2.3. Data Analysis Strategy

To establish preliminary evidence for the reliability and validity of the Well-Lit 6 we used several approaches. First, to determine discriminant validity, we used EFA with principal components estimation and oblique rotation to test whether the items of Well-Lit 6 can be empirically differentiated from other variables purported to measure similar, yet conceptually different constructs. Should wellbeing literacy load on its own factor, it provides some evidence of discriminant validity. As recommended by Costello and Osborne [57], a value above 0.30 were considered as a cutoff for good factor loadings. Second, to establish convergent validity, we developed a nomological network across the three samples using Pearson’s correlations to evaluate the associations between Well-Lit 6 and its consequences. Third, to establish the incremental validity [58] of wellbeing literacy over similar existing constructs (e.g., emotion regulation, resilience), we used hierarchical stepwise regression analyses that allowed us to control for resilience and emotion regulation as existing wellbeing and illbeing predictors prior to adding wellbeing literacy into the model. Significant changes in *R*^2^ were used to support incremental variance explained over and above these existing constructs. Finally, we estimated Cronbach’s alpha reliability coefficients to determine the internal consistency reliability of the scale.

## 3. Results

### 3.1. Convergent Validity

To examine the convergent validity of the Well-Lit 6, the 6-items were correlated with other variables with which they should theoretically be related. These results are shown in Table 3 and Table 4 for the three samples. Table 3 shows the results for the student sample and Table 4 shows the results for the parent and staff samples. Results generally showed moderate to strong correlations between wellbeing literacy and all other variables, ranging from −0.268 to 0.566 in the student sample, −0.252 to 0.440 in the parent sample, and −0.232 to 0.388 in the staff sample. As anticipated, all correlations were in the expected direction. That is, wellbeing literacy showed positive associations with life satisfaction, positive mental health (emotional, psychological, and social wellbeing), emotion regulation, and resilience. Similarly, results showed negative associations with anxiety, depression, stress, and loneliness. Thus, these results supported hypotheses H1 and H2.

### 3.2. Discriminant Validity

To examine the discriminant validity of the Well-Lit 6, we used exploratory factor analysis with principal component estimation and oblique rotation to examine whether the 6-items loaded on their own factor when alongside a variety of other well-being measures. For space reasons, we present the results of these analyses in our Appendix A (See Appendix A). Results showed the six items loaded on their own separate single factor when included alongside other established well-being scales. The single factor solution explained 56.59%, 69.74%, and 68.95% variance in the student, parent and staff samples, respectively. The factor loadings of the six items ranged from 0.675 to 0.807 in the student sample, 0.786 to 0.901 in the parent sample, and 0.806 to 0.862 in the staff sample. This provides preliminary evidence of discriminant validity, supporting H3.

### 3.3. Incremental Validity

The incremental validity tests were employed to examine the additive value of wellbeing literacy over and above existing wellbeing related predictors, and are shown in Table 5. As shown in the table, life satisfaction, depression, anxiety, stress, and loneliness were used as the dependent variables, and demographics, emotion regulation, and resilience were hierarchically organized predictors that were added in a stepwise fashion to observe incremental changes in *R*^2^. Thus, age and gender were added first into the model (step 1), after which emotion regulation and resilience were added (step 2), and finally, wellbeing literacy was added in the final stage (step 3). As shown in Table 5, across all dependent variables, the models indicate significant change in *R*^2^ after adding wellbeing literacy, with the largest observed change for life satisfaction: Δ*R*^2^ = 0.027 (*p* < 0.001); β = 0.209 (*p* < 0.001). This suggests wellbeing literacy has incremental validity in predicting life satisfaction over resilience and emotion regulation. Similarly, using the same procedure, wellbeing literacy displayed significant incremental benefit in predicting variance in anxiety, depression, stress, and loneliness over and above demographics, resilience, and emotion regulation, although the Δ*R*^2^ values were notably smaller, ranging from 0.003 to 0.006. Taken together, these results lend support to H4. 

### 3.4. Supplemental Analyses

#### Internal Consistency Reliability

We examined the reliability of the Well-Lit 6 using internal consistency reliability (Cronbach’s α). As shown in Table 3 and Table 4, Cronbach’s alpha values were α = 0.84 for the student sample, and α = 0.91 in both the parent and staff samples. These results indicate acceptable internal consistency for the Well-Lit 6 [59]. 

## 4. Discussion

The aim of the present study was to investigate the construct of wellbeing literacy in educational settings, establish preliminary evidence for a new 6-item scale of wellbeing literacy (Well-Lit 6), and to test whether wellbeing literacy is empirically distinguishable from wellbeing, illbeing, and their predictors. Additionally, we examined whether wellbeing literacy explained additive variance in wellbeing and illbeing over and above emotion regulation and resilience. Supporting H1, wellbeing literacy was positively associated with wellbeing, including life satisfaction, emotional wellbeing, psychological wellbeing, and social wellbeing. Results also showed wellbeing literacy was negatively associated with illbeing, including depression, anxiety, stress, and loneliness (supporting H2). Wellbeing literacy was empirically separable from other wellbeing and illbeing related scales, loading on its own factor (H3), and explained additive variance wellbeing and illbeing after controlling for emotion regulation and resilience (H4). We take these results to offer preliminary evidence for the validity of the Well-Lit 6 as a short measure of wellbeing literacy.

### 4.1. Study Contributions

Our study provides preliminary evidence that the new construct of wellbeing literacy differs from existing variables across various domains of wellbeing and illbeing. We also showed that wellbeing literacy explained incremental variance in wellbeing and illbeing after controlling for emotion regulation and resilience. These results provide preliminary evidence that wellbeing literacy is a unique construct that offers predictive validity in wellbeing and illbeing outcomes beyond existing constructs. 

Our study provides evidence for the first empirical scale of the new construct of wellbeing literacy and we expect that it can be used as a preliminary measure for future research and practice. In particular, the Well-Lit 6 offers researchers a path to establish empirical literature on wellbeing literacy, providing a means to establish evidence for its antecedents, correlates, and consequences. Moreover, it will allow researchers to evaluate whether wellbeing literacy mediates and/or moderates relations between interventions designed to enhance wellbeing and wellbeing outcomes, as has been previously proposed [19]. While this was beyond the scope of the present study, the Well-Lit 6 offers a means to test such research questions.

Our regression analyses demonstrated that wellbeing literacy yielded smaller relations with illbeing than wellbeing related factors. There has been a long discussion about whether wellbeing and illbeing are bipolar or orthogonal constructs [60,61], with some evidence suggesting that while wellbeing and illbeing are relatively independent, they demonstrate moderate correlations [62,63]. Huta and Hawley, for example, found that strengths contribute more to positive wellbeing such as life satisfaction, positive affect, vitality, meaning, and elevating experience, but vulnerabilities related more to illbeing, including negative affect and depression symptoms [63]. Our results share similarities to this literature, showing that wellbeing literacy may contribute more to wellbeing than illbeing. Further research is needed to replicate and explore this.

As discussed, existing health literacies are limited as they are focused almost exclusively on physical health (health literacy), and mental illness (mental health literacy). They therefore do not capture the positive side of mental health and may offer a limited means to enhance public health. Wellbeing literacy differs insofar as it is focused more on human capability and growth potentials, therefore providing a means to advance the positive side of mental health. We expect this difference will offer practitioners at all levels of public health a promising means to improve human welfare beyond just relieving physical or mental health symptoms. Further, wellbeing literacy would offer a model for wellbeing science and have positive implications for practices in the domains of work and family [64].

### 4.2. Limitations and Future Research Directions

In general, the new scale of wellbeing literacy (Well-Lit-6) demonstrated reliability and construct validity. These findings support the construct of wellbeing literacy and provide an empirical base for further investigation. However, it should be acknowledged that the present study has several limitations. First, the Well-Lit 6 was required to be a short scale to gain access to the broad education-based populations. As a result, it was not possible to develop and validate separate measurements for its sub-components that reflect its necessary conditions. Thus, future research should work to develop more comprehensive measures of wellbeing literacy that incorporate these dimensions.

A related limitation is that the data were from an existing dataset that was not aimed specifically for research on wellbeing literacy. Hence, the variables measured in the student sample did not precisely correspond to those used in the parent and staff samples. This meant that all analyses of incremental validity were performed only in the student sample and need to be replicated. Moreover, our tests for incremental validity only covered the most central wellbeing predictors in the literature (emotion regulation and resilience). Future research should further test the incremental validity of wellbeing literacy beyond other predictor variables (e.g., hope or strength use) to evaluate the additive value of this new construct.

Finally, all measures are self-reported, which may produce common method bias [65], resulting in inflation in correlations and regression coefficients. Future research should seek to overcome common method variance by using correlating wellbeing literacy with objective indicators of wellbeing (e.g., biological or socio-economic indicators [66,67], or by using a time lag between measurements [65]. Such an approach may also help to more clearly establish the causal direction of the proposed relations. 

## 5. Conclusions

The present study evaluated the Well-Lit 6 as a new 6-item scale of wellbeing literacy. We found that wellbeing literacy demonstrated convergent validity by relating in expected directions with wellbeing and illbeing. The scale demonstrated discriminant validity by loading on its own factor when incorporated into a broader EFA model containing several indicators of wellbeing and illbeing. Finally, the scale demonstrated incremental validity in wellbeing and illbeing after controlling for emotion regulation and resilience. Taken together, we accept these data to suggest that the Well-Lit 6 offers a preliminary yet valid measure of the new construct of wellbeing literacy.

## Figures and Tables

**Table 1 ijerph-18-01485-t001:** Items of Well-Lit 6.

	Item	Interpretation
1	I have many words I can think of to communicate about wellbeing	“Vocabulary” (from the definition of wellbeing literacy)
2	I know a lot about wellbeing	“Knowledge” (from the definition)
3	I know how to improve my wellbeing	“Skills” (from the definition)
4	I have the skills to understand information about wellbeing	“Comprehending” in the model of literacy
5	I have the skills to express myself about wellbeing	“Composing” in the model of literacy
6	I can communicate about wellbeing in multiple ways (e.g., writing, listening, drawing) to suit the needs of my audience	Reflecting the multimodal and context-relevant features in the model of literacy

**Table 2 ijerph-18-01485-t002:** Summary of survey measures.

Constructs Measured	Scale Used	Student Survey	Staff Survey	Parent Survey
Wellbeing literacy	Newly developed 6-item self-report (Well-Lit 6) scale	✓	✓	✓
**Wellbeing**				
Emotional wellbeing	Mental Health Continuum-Short Form (MHC-SF) emotional wellbeing sub-scale with three items		✓	✓
Social wellbeing	MHC-SF social wellbeing sub-scale with five items		✓	✓
Psychological wellbeing	MHC-SF psychological wellbeing sub-scale with six items		✓	✓
Life satisfaction	Adapted 2-item scale from the Student’s Life Satisfaction Scale	✓		
**Illbeing**				
Depression	Adapted 2-item from the Patient Health Questionnaire-4 (PHQ-4)	✓	✓	✓
Anxiety	Adapted 2-item from the PHQ-4	✓	✓	✓
Stress	Adapted 2-item from the Perceived Stress Scale	✓	✓	✓
Loneliness	Internally developed 2-item self-report scale (with one more item for the student sample)	✓	✓	✓
**Wellbeing or Illbeing predictors**				
ER—Reappraisal	Adapted 4-item from the Emotion Regulation Questionnaire (ERQ)	✓		
ER—Suppression	Adapted 2-item from the ERQ	✓		
ER—Rumination	Adapted 2-item from the Cognitive Emotion Regulation Questionnaire	✓		
Resilience	Adapted 5-item from the Brief Resilience Scale	✓		

Note. ER = Emotion Regulation.

**Table 3 ijerph-18-01485-t003:** Correlations between the study variables in the student sample.

		*alpha*	*M*	*SD*	1	2	3	4	5	6	7	8	9
1	Wellbeing literacy	0.84	4.60	1.11									
2	Life Satisfaction	0.90	15.08	4.39	0.513 **								
3	ER—Reappraisal	0.81	18.14	4.80	0.566 **	0.554 **							
4	ER—Rumination	0.70	10.35	2.65	0.089 **	−0.009	−0.018						
5	ER—Suppression	0.69	9.71	2.85	−0.115 **	−0.174 **	−0.077 **	0.296 **					
6	Resilience	0.71	21.40	5.59	0.481 **	0.580 **	0.641 **	−0.193 **	−0.158 **				
7	Depression	0.79	7.48	3.27	−0.333 **	−0.621 **	−0.411 **	0.273 **	0.346 **	−0.550 **			
8	Anxiety	0.80	9.45	3.32	−0.142 **	−0.333 **	−0.305 **	0.390 **	0.282 **	−0.541 **	0.574 **		
9	Stress	0.52	8.52	3.06	−0.209 **	−0.463 **	−0.376 **	0.368 **	0.303 **	−0.586 **	0.674 **	0.752 **	
10	Loneliness	0.76	10.94	4.53	−0.268 **	−0.499 **	−0.315 **	0.265 **	0.362 **	−0.443 **	0.711 **	0.497 **	0.574 **

Note. ER = Emotion Regulation. ** *p* < 0.01.

**Table 4 ijerph-18-01485-t004:** Correlations between the study variables in the parent and staff samples.

		*alpha*	*M*	*SD*	1	2	3	4	5	6	7	8
1	Wellbeing literacy	0.91 (0.91)	5.68 (6.00)	0.92 (0.75)		0.254 **	0.374 **	0.388 **	−0.232 **	−0.131 *	−0.172 **	−0.227 **
2	EWB	0.89 (0.90)	8.69 (9.07)	2.70 (2.57)	0.325 **		0.661 **	0.744 **	−0.623 **	−0.481 **	−0.519 **	−0.574 **
3	SWB	0.86 (0.86)	10.54 (12.41)	5.22 (4.85)	0.337 **	0.653 **		0.755 **	−0.522 **	−0.382 **	−0.424 **	−0.533 **
4	PWB	0.87 (0.87)	15.79 (16.56)	5.59 (5.31)	0.440 **	0.728 **	0.712		−0.577 **	−0.487 **	−0.509 **	−0.583 **
5	Depression	0.93 (0.86)	2.92 (2.77)	1.64 (1.54)	−0.252 **	−0.630 **	−0.480	−0.550 **		0.618 **	0.650 **	0.677 **
6	Anxiety	0.87 (0.80)	4.01 (3.97)	1.73 (1.77)	−0.167 **	−0.457 **	−0.419	−0.428 **	0.627 **		0.791 **	0.569 **
7	Stress	0.77 (0.74)	3.47 (3.36)	1.57 (1.58)	−0.247 **	−0.531 **	−0.420 **	−0.536 **	0.675 **	0.754 **		0.618 **
8	Loneliness	0.85 (0.84)	3.14 (2.86)	1.71 (1.61)	−0.272 **	−0.579 **	−0.498 **	-0.554 **	0.708 **	0.584 **	0.673 **	

Note. EWB = Emotional Wellbeing, SWB = Social Wellbeing, PWB = Psychological Wellbeing. The numbers out of the brackets and below the diagonal are results for the parent sample, and numbers in the brackets and above the diagonal are results for the staff sample. * *p* < 0.05; ** *p* < 0.01.

**Table 5 ijerph-18-01485-t005:** Results of hierarchical multiple regression analysis.

	Life Satisfaction		Depression		Anxiety		Stress		Loneliness	
Predictor	β	Δ*R*^2^	β	Δ*R*^2^	β	Δ*R*^2^	β	Δ*R*^2^	β	Δ*R*^2^
**Step 1**		0.007 *		0.011 **		0.060 **		0.042 **		0.008 **
Age	−0.079 *		0.097 **		0.103 **		0.092 **		0.088 **	
Gender^a^	0.015		−0.037		−0.223 **		−0.182 **		−0.012	
**Step2**		0.409 **		0.387 **		0.353 **		0.402 **		0.298 **
Age	−0.041 *		0.057 **		0.067 **		0.052 *		0.054 *	
Gender^a^	−0.049 *		0.049 *		−0.129 **		−0.085 **		0.061 **	
ER - Reappraisal	0.290 **		−0.126 **		0.022		−0.046		−0.08 **	
ER - Rumination	0.105 **		0.125 **		0.243 **		0.217 **		0.126 **	
ER - Suppression	−0.121 **		0.235 **		0.132 **		0.157 **		0.267 **	
Resilience	0.400 **		−0.412 **		−0.467 **		−0.476 **		−0.331 **	
**Step 3**		0.027 **		0.003 **		0.006**		0.003 **		0.003 *
Age	−0.038		0.056 **		0.068 **		0.052 **		0.054 *	
Gender ^a^	−0.033		0.044 *		−0.121 **		−0.079 **		0.056 *	
ER - Reappraisal	0.205 **		−0.097 **		−0.019		−0.075 **		−0.054	
ER - Rumination	0.071 **		0.136 **		0.226 **		0.206 **		0.136 **	
ER - Suppression	−0.101 **		0.229 **		0.142 **		0.163 **		0.261 **	
Resilience	0.349 **		−0.395 **		−0.492 **		−0.493 **		−0.315 **	
Wellbeing literacy	0.209 **		−0.069 **		0.101 **		0.070 **		−0.064 *	

Note. ER = Emotion Regulation. ^a^ 0 = female, 1 = male. * *p* < 0.05; ** *p* < 0.01.

## Data Availability

The data presented in this study are available upon reasonable request from the corresponding author. The data are not publicly available due to restrictions of privacy and ethical considerations of the Department of Education and Training Victoria.

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
