# Peer review of "Wellbeing Literacy: Conceptualization, Measurement, and Preliminary Empirical Findings from Students, Parents and School Staff"

_ijerph, 2021, doi:10.3390/ijerph18041485_

Round 1

Reviewer 1 Report

Thank you very much for giving me the opportunity to read your article

Line 76: there is a reference with author's name, it should be deleted and leave the page number

Aims and Hypotheses: Later, in the discussion, the H1s are referenced, so it is important to specify in the manuscript the hypotheses as follows: Hypothese 1(H1)
To make the discussion clearer.

Procedure: it is necessary to specify the code of the ethics committee by which this study was evaluated 

Table 5: to back up some of the significant data of the r-square, since they are very low. Also, table 5 moves. 

Line 367: correct error

Author Response

Thank you very much for reviewing our article. We appreciate your comments and suggestions and they were helpful in improving the paper:

Below are responses and changes to the reviewer’s comments and suggestions:

  • Line 76:there is a reference with author's name, it should be deleted and leave the page number

Response: Corrected.

  • Aims and Hypotheses: Later, in the discussion, the H1s are referenced, so it is important to specify in the manuscript the hypotheses as follows:Hypothese 1(H1)
    To make the discussion clearer.

Response: Made changes as suggested to make it clearer.

  • Procedure: it is necessary to specify the code of the ethics committee by which this study was evaluated 

Response: Added the codes of ethics.

  • Table 5:to back up some of the significant data of the r-square, since they are very low. Also, table 5 moves. 

Response: We have made no change here since we have discussed the smaller ΔR2 with illbeing related factors than wellbeing related factors. Table 5 has been adjusted to ensure it is in the right position.

  • Line 367:correct error

Response: We are unsure of the error that is being referred to in line 365. However, all of the manuscript has been read through with spelling and grammar errors corrected.

Reviewer 2 Report

This is an interesting article as far as the subject matter is concerned, well written although it is important to re-read it in order to make some grammatical revisions that can improve some aspects of the translation.

The sample, which is large, as well as the method and results, are well presented. And I agree with the proposals for improvement, when the authors consider that the test they use needs to be improved and refined.

I therefore suggest that the article is publishable, with minor changes aimed at improving some aspects of translation, easily remedied by a closer reading of the publication.

Author Response

Thank you very much for reviewing our article. We appreciate your comments and suggestions and have re-read the manuscript and corrected the spelling and grammar errors.

Reviewer 3 Report

The study offers an interesting contribution to the conceptualization and distinctive characteristics of wellbeing literacy. This concept constitutes a research area of special relevance in the field of social and educational studies. The concept is satisfactorily addressed and the methodological procedure is solidly applied. The contributions of the study can potentially increase the advances, already available, in this line of research. In any case, it is recommended to deepen in the implications and potentialities of the new wellbeing literacy construct in the context of specific lines of social research.

Author Response

Thank you very much for reviewing our article. We appreciate your comments and suggestions.

  • In any case, it is recommended to deepen in the implications and potentialities of the new wellbeing literacy construct in the context of specific lines of social research.

We offer the following reflection:

Our paper is focused on the empirical aspect of wellbeing literacy. Thus, the discussion of implications was intentionally constructed to relate to the empirical findings of the Well-Lit 6 measurement, rather than the theoretical potentialities of the construct.

In addition, this paper is one in the special issue "The Contribution of Positive Psychology and Wellbeing Literacy". There is a theoretical paper in this special issue discussing the implications and potentialities of the new construct within a broader context of social research. As such, we referred to that article about the implications of wellbeing literacy (Line 424-426).

Reviewer 4 Report

This paper mainly evaluated the human aspects of the concept of wellbeing literacy and empirically test its relationship with wellbeing and illbeing, futher assessing its incremental variance in wellbeing and illbeing, after controlling for existing and well-established predictors of these constructs within education settings.

The introduction is logical and progressive, and the research problem is clear. The article begins by emphasizing the importance of wellbeing, on this basis, the concept of wellbeing literacy is proposed. Then the interconnection between health and education, the specific conceptual connotations of Wellbeing literacy, and the Short Scale of Wellbeing literacy are described in three parts respectively. Finally, the research aims of this paper and the hypotheses are extended. Moreover, the study contributions and limitations of the article are perceived very clearly, providing a guidance for further and more in-depth research.

However, there are still some problems I think need further modification. Firstly, in detail, many spelling and formatting errors need to be revised; for example, misspelling of “understand” in Table1; incorrect format of literature citation in line 401; incorrect usage of definite and indefinite articles in line 442. Secondly, all measures are self-reported, which is highly subjective. Future research should work to develop more comprehensive and objective indicators of wellbeing. Thirdly, due to the limitation of data, the variables measured in the student sample did not precisely correspond with that used in the parent and staff samples, which may need a deeper explanation.

Taken as a whole, this is an innovative paper that consider the deep relationship between wellbeing literacy and wellbeing in the broader context of education from the perspective of wellbeing literacy. It can actually provide good insights for better understanding of wellbeing.

Author Response

Thank you very much for reviewing our article. We appreciate your comments and suggestions and they were helpful in improving the paper:

Below are responses and changes to the reviewer’s comments and suggestions:

  • Firstly, in detail, many spelling and formatting errors need to be revised; for example, misspelling of “understand” in Table1; incorrect format of literature citation in line 401; incorrect usage of definite and indefinite articles in line 442.

Response: We have re-read the manuscript and corrected the spelling, formatting, and grammar errors.

  • Secondly, all measures are self-reported, which is highly subjective. Future research should work to develop more comprehensive and objective indicators of wellbeing.

Response: We agree with you on this point, and this has been discussed it in the limitations. We therefore have not made any further changes in this regard.

  • Thirdly, due to the limitation of data, the variables measured in the student sample did not precisely correspond with that used in the parent and staff samples, which may need a deeper explanation.

Response: Thank you for point this. We have added the following explanations in the manuscript: “The parents and school staff took fewer measures than students because 1) some items in the survey battery are designed for students, and 2) there was less available time for parents and staff to do the measurements.” (Line 252-254)